# Mental Health Correlates of Autism Spectrum Disorder in Gender Diverse Young People: Evidence from a Specialised Child and Adolescent Gender Clinic in Australia

**DOI:** 10.3390/jcm8101503

**Published:** 2019-09-20

**Authors:** Simone Mahfouda, Christina Panos, Andrew J.O. Whitehouse, Cati S. Thomas, Murray Maybery, Penelope Strauss, Florian D. Zepf, Amanda O’Donovan, Hans-Willem van Hall, Liz A. Saunders, Julia K. Moore, Ashleigh Lin

**Affiliations:** 1Telethon Kids Institute, The University of Western Australia, Perth 6008, AustraliaCati.thomas@telethonkids.org.au (C.S.T.); penelope.strauss@telethonkids.org.au (P.S.); florian.zepf@med.uni-jena.de (F.D.Z.); Ashleigh.Lin@telethonkids.org.au (A.L.); 2School of Psychological Science, Faculty of Science, The University of Western Australia, Perth 6008, Australia; murray.maybery@uwa.edu.au; 3Centre and Discipline of Child and Adolescent Psychiatry, Psychosomatics and Psychotherapy, Division of Psychiatry and Clinical Neurosciences and Division of Paediatrics and Child Health, School of Medicine, Faculty of Health and Medical Sciences, The University of Western Australia, Perth 6008, Australia; julie.moore@health.wa.gov.au; 4The Gender Diversity Service, Child and Adolescent Mental Health Service, Child and Adolescent Health Service, Perth Children’s Hospital, Nedlands 6009, Australia; Hans-Willem.VanHall@health.wa.gov.au (H.-W.v.H.); Elizabeth.Saunders@health.wa.gov.au (L.A.S.); 5School of Population and Global Health, Faculty of Health and Medical Sciences, The University of Western Australia, Perth 6008, Australia; 6Department of Child and Adolescent Psychiatry, Psychosomatic Medicine and Psychotherapy, Jena University Hospital, Friedrich Schiller University, 07743 Jena, Germany; 7Discipline of Psychology, College of Science, Health, Engineering and Education, Murdoch University, Perth 6150, Australia; ccp94@hotmail.com (C.P.); A.ODonovan@murdoch.edu.au (A.O.); 8The School of Human Sciences (Exercise and Sports Science), The University of Western Australia, Perth 6008, Australia

**Keywords:** gender diverse, transgender, autism, ASD

## Abstract

Research suggests an overrepresentation of autism spectrum diagnoses (ASD) or autistic traits in gender diverse samples, particularly in children and adolescents. Using data from the GENTLE (GENder identiTy Longitudinal Experience) Cohort at the Gender Diversity Service at the Perth Children’s Hospital, the primary objective of the current retrospective chart review was to explore psychopathology and quality of life in gender diverse children with co-occurring ASD relative to gender diverse children and adolescents without ASD. The Social Responsiveness Scale (Second Edition) generates a *Diagnostic and Statistical Manual of Mental Disorders* (DSM-5) score indicating a likely clinical ASD diagnosis, which was used to partition participants into two groups (indicated ASD, *n* = 19) (no ASD indicated, *n* = 60). Indicated ASD was far higher than would be expected compared to general population estimates. Indicated ASD on the Social Responsiveness Scale 2 (SRS 2) was also a significant predictor of Internalising behaviours (Anxious/Depressed, Withdrawn/Depressed, Somatic Complaints, Thought Problems subscales) on the Youth Self Report. Indicated ASD was also a significant predictor of scores on all subscales of the Paediatric Quality of Life Inventory. The current findings indicate that gender diverse children and adolescents with indicated ASD comprise an especially vulnerable group that are at marked risk of mental health difficulties, particularly internalising disorders, and poor quality of life outcomes. Services working with gender diverse young people should screen for ASD, and also provide pathways to appropriate care for the commonly associated mental health difficulties.

## 1. Introduction 

The past decade has seen increasing clinical and empirical interest in the co-occurrence of gender diversity and autism spectrum disorders (ASD). Gender diversity is an umbrella term used to describe an evolving range of labels (e.g., non-binary, transgender, agender) that individuals may apply when their gender identity, expression or behaviours do not conform to the norms and stereotypes of their sex assigned at birth) [1]. While clinically significant distress associated with this incongruence is referred to as gender dysphoria [2], the spectrum of gender diversity is increasingly recognised as a normal part of human variation. ASD, according to the Diagnostic and Statistical Manual of Mental Disorders (DSM-5), comprises a group of early-onset neurodevelopmental disorders characterised by persistent deficits in social communication skills across multiple contexts, in addition to restricted and/or repetitive interests and behaviours [2]. The broader spectrum of ASD traits is also recognised as part of normal human diversity [3].

Research suggests an overrepresentation of ASD diagnoses or autistic traits in gender diverse samples, particularly in children and adolescents [4,5,6,7,8], compared to the general population [9,10]. For example, data from Trans Pathways, the largest study conducted in Australian trans and gender-diverse young people (*n* = 859; mean age = 19.4), found that 22.5% of the sample reported having received a formal ASD diagnosis, while more than one third (35.2%) warranted further diagnostic assessments to clarify whether ASD was present [11]. Causal underpinnings of this co-occurrence remain poorly understood, and while theories have been postulated (see Van Der Miesen & Colleagues for a review of underlying hypotheses [12]), a robust evidence base is lacking. Gender diverse behaviours and interests in children and adolescents with ASD may be dismissed by families and clinicians as issues that are part of the ASD phenotype (i.e., restricted, repetitive interests that are unusual in focus). However, the clinical consensus is that in most cases, gender diverse identities and behaviours are not secondary to ASD but co-occur as an aspect of personal identity [13,14,15,16]. 

There has recently been a surge in referrals to specialised child and adolescent gender clinics generally, both within Australia and internationally [1,17,18,19,20]. Gender diverse young people and their families present to these services for a number of reasons, including multidisciplinary assessment and support, but often to explore eligibility and wishes for hormonal interventions, including puberty suppression (e.g., gonadotropin-releasing hormone agonists) and/or gender-affirming medications (e.g., testosterone for those wishing for masculinising effects, and oestrogen for those wishing for feminising effects) [21,22]. 

One study estimated that approximately 7.8% of clinic-referred children and adolescents met formal diagnostic criteria for ASD, as assessed using the Diagnostic Interview for Social and Communication Disorders-10th Revision [4]. However, rates of ASD amongst child and adolescent gender clinics are thought to be higher due to the complexity of diagnosis in many cases [13]. ASD diagnoses or traits do not preclude eligibility for gender-affirming interventions when appropriate. This co-occurrence, however, is often associated with significant diagnostic and treatment challenges, due to ASD-related complexities that may be present, such as difficulties with communication, cognitive flexibility, organization and/or future planning [23]. 

In light of these considerations, it is important that support and interventions for individuals with co-occurring gender diversity and ASD or high autistic traits are tailored in accordance with the young person’s needs, taking individual communication and learning style into account. Where the young person is eligible for hormonal intervention, for example, the risks and benefits may need to be presented in a concrete, literal, detailed manner that caters to the individual’s cognitive skills to facilitate informed consent. If hormonal treatment is initiated, it may be helpful to commence at a lower dose and increase the regimen more gradually, particularly if there are ASD-related issues around hypersensitivity to sensory input and/or difficulties with changes in routine [13]. 

Awareness of other co-occurring difficulties and/or disorders that may be present in this subgroup are also critical considerations that can help inform clinical decision-making and improve quality of life outcomes. These include internalising difficulties (e.g., feelings of anxiety and sadness, etc.) and externalising difficulties (e.g., hyperactivity, defiance), which may be clinically significant. To our knowledge, however, there have been no investigations of the psychological profiles or general quality of life of gender diverse children and adolescents with co-occurring ASD relative to gender diverse children and adolescents without ASD. Mental health difficulties, including generalised anxiety disorder, suicidal ideation and intent, and attention-deficit hyperactivity disorder, are significantly more prevalent in young gender diverse [24,25,26] and ASD [27,28,29,30] groups. In addition, while each group is exposed to significant levels of stigma and prejudice, gender diverse young people with ASD experience lower rates of parental support when disclosing concerns around their gender identity, which may exacerbate distress should they wish to transition socially (which might include a change in gender pronouns) or medically (via hormonal interventions) [31]. 

The Gender Diversity Service (GDS), located at the Perth Children’s Hospital, is the only specialised service in Western Australia to offer support and gender-affirming care to gender diverse children and adolescents (under the age of eighteen) and their families. In accordance with Clinical Guidelines for Co-occurring ASD and Gender Dysphoria or Incongruence in Adolescents [13], the GDS screens for ASD. In addition to clinical interview, the clinic uses the Social Responsiveness Scale 2 (SRS-2) [32], a 65-item measure that identifies the presence and severity of ASD-specific social impairments and restricted interests and repetitive behaviours. The SRS-2 also generates a DSM-5 severity score, thereby providing a metric that reflects the conceptualisation of these symptom categories as described in the DSM-5. Two child and adolescent gender clinics have used SRS data to examine the distribution of ASD traits, which indicated that between 44.9% and 54.2% of the respective samples scored outside the ‘normal’ range [16,33]. However, these studies used older versions of the SRS which did not include the DSM-5 algorithm that indicates the likelihood of a clinical ASD diagnosis. 

Drawing from data from the GDS, the primary objective of the current retrospective chart review was to explore the association between ASD, psychopathology, and quality of life. Using the SRS-2, we examined the distribution of autistic traits in the GDS sample, with a focus on individuals scoring in the ‘severe’ range on the DSM-5 subscale as an indicator of ASD. Our secondary aim was to explore the psychological outcomes and quality of life of participants with indicated ASD compared to those who were not within the ‘severe’ range on the SRS-2 DSM-5 subscale (no ASD indicated). In light of existing research [4,5,6,7,8], we hypothesised that a significantly larger proportion of participants would have indicated ASD compared to estimates from the general population. Similarly, given that ASD and gender diversity are independent risk factors for psychopathology and lower quality of life outcomes, it was hypothesised that the indicated ASD group would experience significantly poorer mental health indices and quality of life, compared to the no ASD indicated group. 

## 2. Methods

### 2.1. Procedure

The GDS, located at Perth Children’s Hospital in Perth, Western Australia, is a dedicated tertiary service working with gender diverse children and adolescents and their families. It is an evidence-based, best-practice tertiary Tier-4 clinical service which provides information, consultation, assessment, support, and access to puberty suppression and gender-affirming oestrogen or testosterone hormonal interventions for young people under 18. One hundred and twenty two prospective participant families were contacted by the Research Officer, either via phone, letter, or in person prior to individual clinical appointments at the GDS (for current patients). The purpose of this initial contact was to gauge interest in their being part of the GENder identiTy Longitudinal Experience (GENTLE) Cohort, of which the sample for the current study is derived. The GENTLE project seeks to examine medical and psychological data in patients who have sought intervention and assessment at the GDS for research purposes, and hence, all patients (past, current and new) are eligible to participate. Families expressing their interest were booked in to discuss the prospect of consenting their hospital data. Seventeen participants declined to participate, and 1 decided to withdraw from the Cohort. For the present study, participants were consented between November 2017 and June 2019. Information from the appointment closest to intake was collated using the participant’s medical file and mental health files. All data were de-identified, and the limits of confidentiality were discussed with each participant. Informed written consent was obtained from each participant before their clinical information could be accessed for analyses. The Child and Adolescent Health Service Human Research Ethics Committee have approved the GENTLE Cohort (#RGS320).

### 2.2. Participants

The final sample was comprised of 104 participants, consisting of all the patients who had given consent to GENTLE participation and for whom there was a completed SRS-2 parent report measure. The average age of the sample at referral was 14.62 (*SD* = 1.72); 25 were birth assigned male (24.0%) and 79 were birth assigned female (76.0%), which is representative of the typical birth assigned sex (BAS) ratio of referrals to the service. The stated gender identities of the participants were male (*n* = 71), female (*n* = 23), non-binary/agender (*n* = 6), and ‘not specified’ (*n* = 4). 

### 2.3. Measures

#### 2.3.1. Social Responsiveness Scale 2 (SRS-2)

The SRS-2 is a parent-report screening tool that is designed to ascertain the presence and severity of core behaviours that characterise ASD [32]. It is comprised of 65 questions rated on a 4-point Likert scale (from 0 = never true to 3 = almost always true) and identifies social impairments in children across five subscales: Social Awareness, Social Cognition, Social Communication, Social Motivation, and Autistic Mannerisms. It yields a Total score, and a score for each subscale. Notably, the SRS-2 also generates a DSM-5 severity score, derived from combining two additional subscales; Social and Communication Impairments, and Restricted and Repetitive Behaviours. In a community population, the SRS-2 distinguishes between children and adolescents without ASD, and those with clinically significant levels of ASD traits [34] Ratings are expressed as *t*-scores (*M* = 50, *SD* = 10). *t*-scores are classified as being in the ‘normal’ (≤59), ‘mild’ (60–65), ‘moderate’ (66–74) or ‘severe’(≥75) range. While *t*-scores of 60 or higher indicate clinically significant ASD symptoms, *t*-scores of ≥75 are associated with a categorical ASD diagnosis [35], with a sensitivity of 0.85 and a specificity of 0.75 for generating a consensual diagnosis [36]. As the emphasis of the current study was the presence of categorical ASD diagnosis, participants were classified as ‘ASD indicated’ if they had scored in the ‘severe’ range in the DSM-5 subscale, which is consistent with the methodologies employed in other recent empirical studies [37,38]. 

#### 2.3.2. Achenbach Youth Self Report (YSR)

The YSR assesses behavioural and emotional difficulties across multiple domains in adolescents aged 11 to 18 years of age. It is comprised of 118 items rated on a 3-point Likert scale [39]. A higher score indicates elevated levels of behavioural and emotional problems. The YSR specifies cut-offs for borderline and clinical ranges on the following subscales: Anxious/Depressed, Withdrawn/Depressed, Somatic Complaints, Social Problems, Thought Problems, Attention Problems, Rule-Breaking Behaviour, and Aggressive Behaviour. A Total score for all behaviours is yielded, in addition to a total score for Internalising behaviours (Anxious/Depressed, Withdrawn/Depressed, Somatic Complaints, Thought Problems) and Externalising behaviours (Social Problems, Attention Problems, Rule-Breaking Behaviour, Aggressive Behaviour). Where participants had literacy or cognitive difficulties, the YSR was not completed. To ensure that the current sample was representative of the GDS population, participants were not excluded from the study on this basis and the other measures were used in analysis.

#### 2.3.3. Paediatric Quality of Life Inventory (PedsQL)

The PedsQL is a brief standardized instrument that assesses health-related quality of life and adaptive functioning in children and adolescents and utilises both a self-report and parent-rated version of the questionnaire [40]. It consists of 23 items rated on a 5-point Likert scale covering four dimensions of functioning: Physical, Emotional, Social and School Behaviours. A higher score on the PedsQL indicates better health-related quality of life. Each item is reverse-scored and linearly transformed on a continuous scale between 0 and 100 (0 = 100, 1 = 75, 2 = 50, 4 = 0. The PedsQL yields a score for each dimension; a Psychosocial Health Summary Score (comprised of Emotional, Social, and School functioning), a Physical Health Summary Score (comprised of Physical functioning), and a Total Score, calculated by summing all the items. The Total score is calculated by the summing all the items over the number of items answered on all the scales. The child version (8–12 years) and adolescent version (13–17 years) were both used to ensure developmental appropriateness. The wording and content for the items on the child and adolescent forms are similar, but the scales remain separate to allow for developmental differences in cognitive abilities. 

## 3. Results

Of this sample, 77.90% (*n* = 81) reported a history of any mental health problems, deliberate-self harming behaviours and/or suicidal intent or attempts, while 11.50% (*n* = 12) did not report any such history. Mental health data were missing for 10.58% (*n* = 11) of participants. At entry to the GDS, 9.62% (*n* = 10) of the sample reported a formal diagnosis of ASD; this information was available for 10.58% (*n* = 11 of the sample). 

Sample demographics (including sex assigned at birth and gender identity) and descriptive statistics for psychological measures (YSR and PedsQL) are presented in Table 1. Mean scores on the YSR were high, with the mean *t*-score for Internalising and Total behaviours within the clinical range. In this sample, 65.4% (*n* = 68) of participants fell into the clinical range on Internalising behaviours, 14.4% (*n* = 15) fell in the clinical range on Externalising behaviours, and 54.8% (*n* = 57) fell into the clinical range on Total behaviours. 

The proportion of participants falling into the ‘normal’, ‘mild’, ‘moderate’, and ‘severe’ ranges based on SRS-2 scores are presented in Table 2. For SRS-2 Total scores, 51% of the sample fell within the ‘normal’ range. The remaining 49% of the sample were approximately equally distributed across the ‘mild’ (16.3%), ‘moderate’ (14.4%) and ‘severe’ (18.3%) clinical ranges. The sample was similarly distributed on the DSM-5 subscale, with 17.3%, 9.6% and 22.1% of the sample being spread across the ‘mild’, ‘moderate’ and ‘severe’ categories.

Based on the SRS-2 DSM-5 subscale, the sample was partitioned into two groups: participants who fell in the ‘severe’ range (*n* = 23; 22.1%) (indicated ASD), and participants who did not fall within the ‘severe’ range, i.e., were ‘normal’, ‘mild’ or ‘moderate’ (*n* = 77; 74.0%) (no ASD indicated). Demographic and descriptive statistics for these two groups are presented in Table 3. Chi-square analyses indicated no significant group differences on sex assigned at birth or gender. There were also no statistically significant differences between groups with respect to age at referral (*t* = 1.16 (98), *p* = 0.2) and age at which the SRS-2 was completed in the clinic (*t* = 0.37, (98), *p* = 0.7). 

Descriptive statistics for the YSR and PedsQL for these two groups are presented in Table 3. Linear regressions using indicated ASD vs. no ASD indicated as a predictor of continuous psychopathology outcomes (YSR and PedsQL) were conducted, with sex-assigned at birth, time between assessments and age at SRS-2 administration as covariates. These findings are presented in Table 4. Indicated/not indicated ASD was significantly associated with all subscales of the PedsQL, and all YSR subscales except Externalising behaviours, suggesting higher levels of mental health difficulties and poorer quality of life among these children and adolescents with indicated ASD.

We also explored whether indicated/not indicated ASD was predictive of clinically significant YSR scores. Of those participants in the indicated ASD group, 78.4% (*n* = 18) also fell into the clinical range on Internalising behaviours, 13.0% (*n* = 3) fell into the clinical range on Externalising behaviours, and 82.6% (*n* = 19) fell into the clinical range on Total behaviours. Of the participants with no ASD indicated, 61.0% (*n* = 47) fell into the clinical range on Internalising behaviours, 15.6% (*n* = 12) fell into the clinical range on Externalising behaviours, and 46.8% (*n* = 36) fell into the clinical range on Total behaviours. Binary logistic regression analyses were conducted, with membership in the clinical range on the YSR behaviours as dependent variables and indicated/not indicated ASD as the predictor (covarying for BAS, time between assessments and the age the SRS was administered). Participants with indicated ASD were as likely to fall into the clinical range for the YSR Internalising behaviours (Exp(B) = 2.91 (95% *CI* = 0.87, 9.74, *p* = 0.08)) and YSR Externalising behaviours (Exp(B) = 0.84 (95% *CI* = 0.20, 3.31, *p* = 0.80)) as those with no ASD indicated. For YSR Total behaviours, the odds of falling into the clinical range were significantly higher for participants with indicated ASD (Exp (B) = 7.77 [95% *CI* = 2.04, 29.56], *p* = 0.003). 

## 4. Discussion

Emerging research suggests an overrepresentation of ASD diagnoses and autistic traits in gender diverse children and adolescents [4,5,6,7,8] compared to general population prevalence estimates [41,42]. To our knowledge, there have been no empirical investigations examining measures of psychological symptoms and function, and quality of life, in gender diverse young people with ASD. Using data from the GENTLE Cohort from the Gender Diversity Service at Perth Children’s Hospital, the primary objective of the current retrospective chart review was to explore the association between ASD, psychopathology, and quality of life in gender diverse children and adolescents. 

### 4.1. Autistic Traits in Clinic-Referred Gender Diverse Children and Adolescents 

Our first aim was to examine levels of autistic traits in the sample, with a particular focus on participants scoring in the ‘severe’ range on the SRS-2 DSM-5 subscale ≥75 as an indicator of ASD, which closely aligns with the DSM-5 diagnostic criteria for ASD [32]. In view of existing research regarding high rates of ASD and autistic traits in children and adolescents presenting to specialised gender clinics [4,5,6,7,8], we hypothesised that a larger proportion of the participants would meet the criteria for ASD compared to estimates from the general population. The results of the current study are consistent with this hypothesis, with indicated ASD present in 22.1% (*n* = 23) of the sample. Although prevalence cannot be determined using SRS-2 scores, this estimate is markedly higher than children and adolescents thought to have ASD in the general population, with estimates ranging between 1.7% and 2.47% [41,42]. 

Notably, 9.62% (*n* = 10) of the sample reported a formal diagnosis of ASD at the time of clinical intake, which is also substantially higher than general population prevalence estimates. The discrepancy between indicated ASD on the SRS-2 (*n* = 23) and formally diagnosed ASD (*n* = 10) may be partially attributed to the retrospective design of the current study, as this information was missing from the clinical notes of some participants (10.58%; *n* = 11), and it is possible that some of these participants had an ASD diagnosis. Not all young people who may have ASD in Western Australia have had access to specialist ASD diagnosis services. This may be because the diagnosis is not suspected until adolescence for some young people. For others, it may be because of barriers such as waiting lists at public services, substantial financial cost at private services, geographical remoteness, or family reluctance to pursue assessment. Another explanation could be that ASD traits in gender diverse children and adolescents may be attributed to the wide spectrum of gender diverse behaviours and expressions, which could result in fewer clinical queries of ASD in these young people.

In addition, given that a substantial proportion of the sample were birth-assigned females (76.0%, *n* = 79), another explanation for these findings could be attributed to the effects of BAS on ASD diagnosis. Drawing from the broader ASD literature, it is well-established that females are much less likely to be diagnosed with ASD than males, even when symptoms are equally severe [43], with a reported male-to-female ratio of approximately 4:1 [44]. Females with ASD are thought to be better at ‘camouflaging’ their symptoms through the use of compensatory strategies to mitigate communication and social difficulties [45,46]. Very little is currently understood about how gender variance influences ASD expression and clinical identification, but it is possible that gender diverse BAS may use similar compensatory strategies. However, it is notable that 31.67% of birth-assigned females in the current sample had indicated ASD (19/60), in contrast to 23.5% in birth-assigned males (4/17). These data suggest that ASD might be more prevalent in BAS who are gender diverse, or that the birth-assigned male-to-female ratios of ASD in this group might be closer to 1:1. It is difficult to draw definitive conclusions based on the small sample size, and further studies are warranted to explore this.

It is noteworthy to mention that an additional 26.9% of participants fell into the ‘mild’ (17.3%, *n* = 18) and ‘moderate’ (9.6%; *n* = 10) ranges for the DSM-5 subscale on the SRS-2. Including participants that fell into the ‘severe’ range, these data indicate that almost half of this gender diverse sample have some degree of ASD traits. Although previous studies have used an older version of the SRS, which does not generate a DSM-5 subscale score, Total SRS scores from these studies indicate that between 44.9% and 54.2% of clinic-referred gender diverse children and adolescents fall within this spectrum of scores [16,33], which aligns with our findings.

### 4.2. Associations between ASD, Psychopathology and Quality of Life

Our secondary aim was to explore psychological outcomes and quality of life in the group with indicated ASD compared to the participants with no ASD indicated. As ASD and gender diversity are both independent risk factors for psychopathology and lower quality of life outcomes, it was hypothesised that the ASD group would experience significantly poorer mental health (i.e., elevated YSR scores) and quality of life (i.e., lower PedsQL scores) than participants without ASD. With respect to YSR scores, this hypothesis was partially supported; we demonstrated that indicated ASD was a significant predictor of Internalising behaviours. Indicated ASD was also found to be a significant predictor of overall behavioural problems, whereby participants from this group were almost 8 (7.77) times more likely to fall within the clinical range for the Total subscale of the YSR than those with no ASD indicated. There were no group differences on Externalising behaviours, which suggests that ASD expression in gender diverse populations may be less likely to be associated with clinical externalising problems.

It should be noted that both groups had clinically elevated mean scores for the Internalising scale of the YSR (indicated ASD [*M* = 74.00, *SD* = 13.72]; no ASD indicated [*M* = 64.99, *SD* = 12.25]). This is consistent with the established finding of high rates of psychopathology in gender diverse young people, including internalising psychiatric disorders (e.g., mood disorders, anxiety disorders) [8,24,26]. Collectively, however, the current findings support the notion that clinic-referred gender diverse children and adolescents with co-occurring ASD comprise an especially vulnerable group that are at marked risk of mental health difficulties, particularly internalising psychopathology. 

Indicated ASD was also a significant predictor of scores on all subscales on the PedsQL. This suggests that gender diverse participants with ASD have a reduced quality of life across their physical health, social and emotional wellbeing, and school functioning compared to gender diverse participants without indicated ASD. These findings complement studies that indicate high levels of stigma and prejudice in gender diverse young people with co-occurring ASD, who receive less parental support when disclosing concerns about gender identity [31]. The finding of reduced quality of life indicators in this high-risk group underscore the importance of providing psychosocial support to gender diverse young people with co-occurring ASD and their families. 

### 4.3. Implications

The current investigation is novel in several respects. To our knowledge, this is the first study to estimate the proportion of children and adolescents referred to an Australian gender clinic with ‘indicated ASD’, as indexed by a brief screening measure. The distribution of ASD traits reported here are similar to those reported from other countries that used the SRS in child and adolescent gender clinics [16,33]. These findings highlight the similarities between clinic-referred gender diverse cohorts internationally, an area which has been relatively unexplored. Given the high rates of ASD and autistic traits in these samples, it is pivotal that there is sufficient education amongst health providers working with gender diverse young people, particularly familiarity with current guidelines around this co-occurrence [13]. The current study is also the first to examine the associations between ASD, psychopathology and quality of life in a clinically derived gender diverse sample. Awareness of difficulties that are common in this subgroup, particularly internalising problems, are also important to inform thorough assessment and personalised care planning, including access to evidence-based mental health care, which can sit alongside gender-affirming medical treatment to support overall wellbeing.

The high proportion of the current sample with significant autistic traits relative to self-reported diagnoses indicates that there may be individuals who have not previously been identified. As the GDS does not have capacity and resources to formally diagnose ASD, the inclusion of the SRS-2 as a brief screening measure for suspected ASD is an important addition to our specialised child and adolescent gender clinic assessment pathway, and aligns with current guidelines [13]. Although it is not a diagnostic measure, it is effective and convenient in in flagging potential ASD, which may, in turn, help to inform clinical assessment and treatment decisions, as well as prompting referral for formal ASD assessment. 

### 4.4. Limitations

A limitation of the current study concerns the methodology of evaluating ASD diagnosis. Although the SRS-2 has strength in being aligned with the DSM-5 criteria for ASD, it is a screening tool for the presence and severity of autism traits and an indication of a potential diagnosis only. It is therefore possible that individuals who scored in the ‘moderate’ or even ‘mild’ range of the SRS-2 may meet diagnostic criteria for ASD if formally assessed. The SRS-2 is a parent report instrument, with no youth self-report component available, and no structured direct clinical assessment of autism spectrum characteristics. 

It is also necessary to acknowledge that the individuals referred to the GDS are not representative of all gender diverse children and adolescents in Australia, in that the vast majority of the young people referred to the GDS have some degree of support and acceptance from at least one parent or guardian to facilitate referral, and are involved in the ongoing assessment and treatment. Gender diverse youth who are estranged from family (who have experienced non-acceptance or rejection) are likely to experience more severe mental health distress and may differ from this study population [11]. 

### 4.5. Future Research

There are several avenues for future studies that could be considered. Future researchers should use the findings of this study to inform the development and evaluation of best clinical practice guidelines in Australia. For example, studies could evaluate whether increasing clinicians’ knowledge of ASD in the gender diverse population improves patient care experience and outcomes. This will aim to improve the clinical care provided to gender diverse young people to ensure their specific needs are being catered for. Additionally, future studies should seek to investigate potential causal factors underlying this co-occurrence. Such data can help to improve our understanding of the interaction between gender diversity and ASD, and similarly help to inform the care being provided to this population. 

Finally, it should be noted that although the discourse in the literature regarding the co-occurrence of ASD traits and gender diversity is often one of difficulties, deficits and complexities, our clinical experience working with this group of young people is that great strengths are often evident. These include, but are not limited to creativity, originality and personal integrity, a passion for advocacy, and notable academic, artistic, musical and literary talents. These observations could also guide further research exploring the characteristics of this population.

### 4.6. Conclusions

This study expanded on previous research to replicate the finding of higher levels of autistic traits and ASD in a clinic-referred gender diverse sample in Australia. We found that gender diverse children and adolescents with indicated ASD comprise an especially vulnerable group that are at marked risk of mental health difficulties, particularly internalising disorders, and poor quality of life. These findings should be considered when developing best practice for working with gender diverse young people with ASD to ensure that their health care meets their unique needs.

## Figures and Tables

**Table 1 jcm-08-01503-t001:** Sample demographics, autistic traits, psychopathology and quality of life.

	*N*	%	
BAS			
Female	79	76.0	
Male	25	24.0	
Gender identity			
Female	23	22.1	
Male	71	68.3	
Non-binary/agender	6	5.8	
Other gender identity	4	3.8	
	***N***	***M***	***SD***
Age at referral	104	14.62	1.72
Age at SRS-2 completion	104	15.49	1.67
SRS-2 Social Awareness	104	57.88	12.17
SRS-2 Cognitive	104	55.95	13.04
SRS-2 Communication	104	58.38	13.10
SRS-2 Motor	104	63.39	14.52
SRS-2 Restricted and Repetitive Behaviours	102	62.24	13.76
SRS-2 Social Communication	104	59.83	13.54
SRS-2 DSM-5	100	61.77	14.03
SRS-2 Total score	104	60.68	14.06
YSR Internalising *t*-score	102	67.00	12.83
YSR Externalising *t*-score	102	54.45	10.19
YSR Total *t*-store	102	64.08	10.94
PedsQL Physical standardised score	94	69.40	21.52
PedsQL Emotional standardised score	94	47.43	22.16
PedsQL Social standardised score	94	66.86	23.63
PedsQL School standardised score	93	54.84	21.62
PedsQL Psych standardised score	94	55.95	19.77
PedsQL Total standardised score	94	60.60	19.15

*Note*: BAS = birth-assigned sex, SRS-2 = Social Responsiveness Scale 2, YSR = Youth Self Report, PedsQL = Paediatric Quality of Life Inventory.

**Table 2 jcm-08-01503-t002:** Proportion of sample falling within each range on the SRS-2 total and subscales.

	Within Normal Range		Mild		Moderate		Severe	
	*N*	%	*N*	%	*N*	%	*N*	%
Total score	53	51.0	17	16.3	15	14.4	19	18.3
Social Awareness	63	60.6	15	14.4	18	17.3	8	7.7
Cognitive	70	67.3	10	9.6	14	13.5	10	9.6
Communication	59	56.7	18	17.3	14	13.5	13	12.5
Motor	41	39.4	18	17.3	23	22.1	22	21.2
Restricted and Repetitive Behaviours	46	44.2	21	20.2	13	12.5	22	21.2
Social Communication	56	53.8	14	13.5	17	16.3	17	16.3
DSM-5	49	47.1	18	17.3	10	9.6	23	22.1

*Note*: 2 participants missing Restricted and Repetitive Behaviours score; 4 participants missing DSM-5 score.

**Table 3 jcm-08-01503-t003:** Demographics, autistic traits, psychopathology and quality of life by indicated/not indicated ASD on the SRS-2 DSM-5 subscale.

	Indicated ASD			No ASD Indicated			
	*N*	%		*N*	%		*p*-Value
BAS							0.6
Female	19	82.6		60	77.9		
Male	4	17.4		17	22.1		
Gender identity							0.3
Female	4	17.4		16	20.8		
Male	15	65.2		56	72.7		
Non-binary/agender	2	8.7		4	5.2		
Other gender identity	2	8.7		1	1.3		
	***N***	***M***	***SD***	***N***	***M***	***SD***	
Age at referral	23	14.97	1.18	77	14.49	1.87	0.2
Age at SRS-2 completion	23	15.62	1.42	77	15.47	1.78	0.7
YSR Internalising t-score	22	74.00	13.72	76	64.99	12.25	0.004
YSR Externalising *t*-score	22	56.64	8.56	76	53.80	10.81	0.3
YSR Total *t*-store	22	69.09	12.22	76	62.68	10.46	0.01
PedsQL Physical standardised score	20	53.28	22.91	71	73.70	19.18	<0.001
PedsQL Emotional standardised score	20	38.25	21.11	71	49.84	22.11	0.04
PedsQL Social standardised score	20	50.75	25.41	71	71.13	21.58	0.001
PedsQL School standardised score	20	44.75	22.39	70	57.71	21.14	0.01
PedsQL Psych standardised score	20	45.25	19.52	71	58.79	19.33	0.007
PedsQL Total standardised score	20	47.59	20.13	71	64.07	17.70	0.001

*Note*: BAS = birth-assigned sex, SRS-2 = Social Responsiveness Scale 2, YSR = Youth Self Report, PedsQL = **Paediatric Quality of Life Inventory**.

**Table 4 jcm-08-01503-t004:** Indicated and no indicated ASD on the SRS-2 DSM-5 subscale predicting psychopathology and quality of life.

	*B*	95% *CIs* for B	*p*-Value
**YSR Internalising *t*-score**			
SRS-2 ASD indicator	8.88	2.82, 14.94	0.005
Age at SRS-2 completion	−0.80	−2.34, 0.73	0.30
Birth-assigned sex	−4.61	−10.81, 1.59	0.14
Time between SRS-2 and YSR	−0.37	−4.20, 3.45	0.85
**YSR Externalising *t*-score**			
SRS-2 ASD indicator	2.67	−2.30, 7.64	0.29
Age at SRS-2 completion	−0.66	−1.92, 0.60	0.30
Birth-assigned sex	−4.07	−9.15, 1.02	0.15
Time between SRS-2 and YSR	−1.01	−4.14, 2.13	0.53
**YSR Total t-score**			
SRS-2 ASD indicator	6.24	1.06, 11.42	0.019
Age at SRS-2 completion	−0.83	−2.14, 0.49	0.21
Birth-assigned sex	−5.09	−10.37, 0.21	0.06
Time between SRS-2 and YSR	−0.58	−3.85. 2.60	0.73
**PedsQL Physical standardised score**			
SRS-2 ASD indicator	−19.83	−29.84, −9.81	<0.001
Age at SRS-2 completion	0.74	−1.69, 3.17	0.55
Birth-assigned sex	13.12	2.84, 23.40	0.013
Time between SRS-2 and PedsQL	−1.51	−8.01, 5.00	0.646
**PedsQL Emotional standardised score**			
SRS-2 ASD indicator	−10.92	−21.78, −0.06	0.049
Age at SRS-2 completion	1.36	−1.23, 3.99	0.31
Birth-assigned sex	15.20	4.05, 26.35	0.008
Time between SRS-2 and PedsQL	−1.75	−8.81, 5.30	0.62
**PedsQL Social standardised score**			
SRS-2 ASD indicator	−21.75	−32.98, −10.52	<0.001
Age at SRS-2 completion	0.007	−2.71, 2.73	0.97
Birth-assigned sex	7.07	−4.46, 18.59	0.23
Time between SRS-2 and PedsQL	−7.79	−15.08, −0.50	0.037
**PedsQL School standardised score**			
SRS-2 ASD indicator	−13.16	−24.10, −2.23	0.019
Age at SRS-2 completion	1.11	−1.55, 3.78	0.41
Birth-assigned sex	8.40	−2.83, 19.63	0.14
Time between SRS-2 and PedsQL	−3.39	−10.49, 3.71	0.35
**PedsQL Psych standardised score**			
SRS-2 ASD indicator	−13.77	−23.41, −4.14	0.006
Age at SRS-2 completion	1.12	−1.22. 3.45	0.344
Birth-assigned sex	11.02	1.13, 20.91	0.029
Time between SRS-2 and PedsQL	−4.18	−10.44, 2.07	0.19
**PedsQL Total standardised score**			
SRS-2 ASD indicator	−16.44	−25.47, −7.41	<0.001
Age at SRS-2 completion	1.00	−1.18, 3.19	0.36
Birth-assigned sex	11.75	2.48, 21.02	0.01
Time between SRS-2 and PedsQL	−3.31	−9.17, 2.55	0.26

*Note*: YSR = Youth Self Report, PedsQL = Paediatric Quality of Life Scale.

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
