# Peer review of "Mental Health Correlates of Autism Spectrum Disorder in Gender Diverse Young People: Evidence from a Specialised Child and Adolescent Gender Clinic in Australia"

_jcm, 2019, doi:10.3390/jcm8101503_

Round 1
Reviewer 1 Report
This paper focuses on the mental health associated with autism spectrum disorder in adolescents with gender diversity. This is an area of clinical importance and a well justified area to study. The paper is generally well written although there are numerous formatting, font and referencing errors. I have provided comments page and line by line below. There also appear to be discrepancies between the table and text. The main concerns are around how ‘indicated ASD’ was determined from the SRS scale which is essentially a screener. It was unclear how the cut point of ‘severe’ was determined to indicate ASD but those in the mild and moderate range were then considered to be ‘ASD not indicated’. This binary way of explaining the adolescents seems problematic. Having ‘traits of ASD’ may be a more accurate way to describe the participants.
Page 1
Line 37. Suggest being more specific around what these internalising and total behaviours were- to be high on ‘total behaviours’ doesn’t mean much from a clinical perspective.
Line 39. Assuming this Total Subscale refers to the Youth Self report here? This was a bit unclear.
Line 43. In what way inform best care practice? This is a bit of an empty statement and would benefit from more detail.
Page 2
Line 59. What do these studies specifically report re: prevalence i.e. what are the numbers? Please provide more detail here.
Line 60. References 9,10 for prevalence are somewhat outdated and not the best data available. Either CDC data (for US) or Australian specific data (May et al, Bent et al) may be more appropriate.
Line 62. Theories are postulated here yet no detail provided. It would be relevant to expand this to include what these suggested theories are.
Line 74. Is this true incidence or prevalence here? Can you comment on these key papers- how did they assess ASD?
Line 89. Specifically what is meant by externalising and internalising difficulties here?
Line 108. There is a much higher prevalence for the tool than when formal clinical diagnosis made (ref 4) in other studies. How do the authors explain this discrepancy?
Page 3
Line 124. The period of time (years) participants were recruited should be described.
Line 124. Were all adolescents that were eligible included in the study? Any loss to follow up or individuals that declined to be part of the study? The inclusion/exclusion criteria were not clearly specified. Were only children diagnosed as gender diverse included or were some children not diagnosed as diverse included in this sample? How representative was the sample of those that were eligible for the study but didn’t participate?
Line 142. The SRS doesn’t claim to distinguish ‘neurotypical’- it primarily looks at ASD traits. I appreciate this word is often used in the field of ASD to indicate non-ASD but the tool can’t tell you if a child is or is not neurotypical in the broader sense e.g. whether they have intellectual disability or other comorbidities. Please make amend this e.g. to ‘non ASD’ or something similar.
The discrepancy in numbers between those diagnosed ASD and those showing severe on SRS is remarkable. It is hard to believe only those missing data were the ones diagnosed with ASD or that this difference is only due to access to assessment given there is national funding for services for ASD in Australia.
YSR- Did the children complete this? What did you do if the child had limited language or intellectual disability, as is common in ASD? One would expect some children with these high levels of severity of ASD to also have these types of comorbidities.
Page 4
Table1: What does BAS refer to? Table 1 needs a legend for all the acronyms.
Line 173-175. The numbers don’t appear to add up in the text and table. There appear to be n=101 in the sample (n=80 history mental illness, n=10 no history and n=11 missing data) yet in the table there are some measures of n=104. For example, biological male/female n is 25+79=104. Please either correct or explain this discrepancy.
It would be helpful to include any other diagnoses or comorbidities in the table that are relevant such as ID which is common in ASD if this is available.
Line 177. Was there an ASD expert/clinician in the team to determine high ASD traits? This sentence does not seem necessary if not backed by data.
Page 6
Table 2 and 3. Labels appear to be bolded in some cases and not in others- it was unclear of this is an accident or intentional.
If the PEDSQL produces continuous standardised scores this should be clearly stated in methods. The reader understood this was only a 0-100 scale as compared to a standardised score. Please explain further how the Likert scales are converted and how the total score is obtained. What is within the normal range/cut off for this scale? Is 50 average and 1 or 2 SD either side? Please add this information to the legend for the reader. Also, is the adolescent and child version compatible for combining results?
Page 196. Please explain justification for having only severe category as 'ASD indicated'. How was this decided? What evidence is there to back this cut point? Moderate score on the SRS does not suggest ‘ASD not indicated’. Please justify the rationale for including individuals with moderate ASD in the ‘non-indicated’ group.
Page 7.
Line 241. This appears to be the first time the >76 cut off is mentioned- this should be in the methods.
Line 280. It is conflicting noting that almost 50% were found to have mild, mod, severe ASD and yet mild and mod were put in the ‘not indicated ASD’ group in the analysis. Suggest sticking to one approach.
Line 314. ‘Estimate prevalence of ASD’- this was not a clinical assessment that can estimate ASD from a parent completed screening tool.
Additional comments
There were numerous issues with font, references and formatting. I haven’t pointed out each one but examples include: font is blue in some areas and not others (page 1,3,4) and there are variable gaps after citations within the manuscript. Inconsistencies in way acronyms written e.g. PEDSQL or PEDS-QL. SRS and SRS-2 (Table 1). Text in tables is centred which is a bit confusing to the eye.
Author Response
Please see attached Word document.
Best wishes,
Simone

Reviewer 2 Report
This manuscript is an excellent addition to the growing body of data regarding the elevated incidence of ASD among TGD youth. I am pleased that you chose to frame this investigation with a specific intent to describe, rather than imply a causal relationship between gender diversity and ASD. My recommendations for the presentation of your data is to align your data charts with right justification, rather than center, and add your p-values to Table 3. There is one typo on page 6, line 215, I think you meant "group" not "grop."
Overall, excellent work.
Author Response

(The authors gave the same response as above.)
